# A biomimetic 2D transistor for audiomorphic computing

Sarbashis Das [1], Akhil Dodda [2] & Saptarshi Das [2,3]

In this article, we introduce a biomimetic audiomorphic device that captures the neurobiological architecture and computational map inside the auditory cortex of barn owl known for its exceptional hunting ability in complete darkness using auditory cues. The device consists of multiple split-gates with nanogaps on a semiconducting $MoS_2$ channel connected to the source/drain contacts for imitating the spatial map of coincidence detector neurons and tunable RC circuits for imitating the interaural time delay neurons following the Jeffress model of sound localization. Furthermore, we use global back-gating capability to demonstrate neuroplasticity to capture behavioral and/or adaptation related changes in the barn owl. Finally, the virtual source model for current transport is combined with finite element COMSOL multiphysics simulations to explain and project the performance of the biomimetic audiomorphic device. We find that the precision of the biomimetic device can supersede the barn owl by orders of magnitude.

[1] Electrical Engineering, Pennsylvania State University, University Park, PA 16802, USA. [2] Engineering Science and Mechanics, Pennsylvania State University, University Park, PA 16802, USA. [3] Material Research Institute, Pennsylvania State University, University Park, PA 16802, USA. Correspondence and requests for materials should be addressed to S.D. (email: sud70@psu.edu) or (email: das.sapt@gmail.com)

Today, the supercomputers have become so powerful that they can easily surpass the performance and capacity of the human brain in processing speed and amount of information storage[1,2]. However, there exists a giant gap in energy consumption and area efficiency between the human brain and the supercomputers with the supercomputers being at least 1,000,000× worse than the human brain[3,4]. The emerging era of neuromorphic computing promises to shrink this gap by deploying artificial neural networks (ANNs)[5]. ANNs mimic the fundamental computing unit of brain i.e., neurons connected to other neurons via synapses. Neuromorphic chips such as TrueNorth[6], Loihi[7], BrainScaleS[8], and SpiNNaker[9] are truly impressive advancements in the area of artificial intelligence, however, these chips will still remain overwhelmingly power hungry when scaled up to the full capacity of human brain which has 100 billion neurons connected via 1 quadrillion synapses operating at a miniscule 20 W power. It remains to be seen how the energy efficiency of neuromorphic computing based on conventional devices is improved especially at a time when all three quintessential aspects of Moore's law of scaling i.e., energy, size, and complexity scaling have practically ended. In this article, we, however, adopt a different approach towards neuromorphic computing by embracing the unique neurobiological architecture and associated computational maps that exist inside the brains of animals with extraordinary sensory capabilities such as the barn owl. The organization of neurons and their synaptic connectivity in these natural super sensors have been delicately tuned over millions of years to achieve evolutionary success for the species in extreme, and resource-constrained environments making sensory information processing at ultra-low power. Therefore, mimicking these super sensory neural architectures and associated algorithms through innovative solid-state devices and circuits might provide an alternative route towards energy efficient neuromorphic computing.

## Results

**Neural map and compute algorithm in barn owl**. Ability to localize sound is an essential survival feature for predators and preys alike explaining why terrestrial vertebrates have two ears so that the interaural time difference (ITD) of low frequency sound waves can be used as a cue to determine the direction of its source[10–12]. Fig. 1a shows how the path difference between the sound waves reaching the two ears is translated into ITD for an acoustic source located at an azimuth angle, $\theta$, for a given head radius ($r_H$) and sound velocity ($v_s$). Figure 1b shows ITD as a function of $\theta$ for different head sizes. Clearly, for typical size of animal heads it is imperative that the auditory information must be processed within hundreds of microseconds using neurons which can fire only once per few milliseconds. Therefore auditory signal processing is a challenging computational task for any animal. Remarkably this problem has been evolutionarily resolved using smart neural algorithms implemented through befitting neurobiological architectures. In this context, barn owls are model auditory systems owing to their extraordinary ability to determine the location of sound with a precision of 1–2 degrees even when hunting in total darkness[13]. In 1948, Lloyd Jeffress published a seminal paper, where he formulated a model that describes how acoustic timing differences are represented as a "place" in an array of nerve cells or in other words how the brain transforms temporal coding into spatial coding[11]. Remarkably, the key assumptions of his model are valid even today. In fact the Jeffress model remains as the cornerstone for the neurophysiological understanding and development of most computational models for sound localization[14,15]. Fig. 1c depicts the Jeffress model with three essential neural components: (1) the time delay neurons, (2) the coincidence detector neurons, and (3) a spatial computational map

that correlates the two. The coincidence neurons fire only when spikes arrive concurrently to the corresponding delay neurons. The length of the delay neurons from the left and right cochlear nuclei is equal up to the points X and Y. However, beyond these points these delay neurons are projected onto coincidence neurons in a ladder-like branching structure that runs in opposite directions. The delay neurons from the right side form a set of collaterals so that the axonal length to the coincidence neuron is longer for cell 7 than for cell 1 and vice versa for the delay neurons from the left side. Due to the finite speed of axonal conduction, these branching patterns constitute a map for the ITDs and hence correlates to the spatial location of the sound source. For example if a sound originates from straight ahead, it will reach the right and left cochlea at the same time, i.e., without any interaural delay (ITD = 0 μs). In this case only neuron 4 will receive coincident inputs since the total axonal path lengths are equal. However, if the sound originates from the left hemisphere, it will reach the left ear earlier than the right ear. In this case, coincidence will occur if the input signal from the left travels a longer path length than that from the right by an amount that exactly offsets the acoustic delay, e.g., at cell 1. Similarly, if the sound originates from the right hemisphere, coincidence will occur e.g., at cell 7. Therefore, the spatial projection of the right and left delay neurons onto the coincidence neurons form a computational map, where each coincidence neuron represents a specific ITD corresponding to an azimuth. Figure 1d shows a simple anatomical drawing of barn owl's brainstem which bears astonishing similarity with the Jeffress model[16,17]. In the barn owl, the axons of secondary nerve fibers from the nucleus magnocellularis (NM) provide the delay lines. These monaural channels originate from left and right cochlear nuclei and converge on binaural tertiary nerve fibers which serve as coincidence detectors in the nucleus laminaris (NL). Neurons in NL only discharge on receiving coincident spikes from their monaural, excitatory afferents. Figure 1e shows the schematics of solid state coincidence detector neuron and delay neuron. We have used a split-gated MoS$_2$ field effect transistor (FET) to achieve the functionality of coincidence detector neuron and tunable RC circuits to mimic the delay neurons. Finally, Fig. 1f shows the schematic of the fully integrated biomimetic audiomorphic device in order to emulate the neural computational map inside the auditory cortex of barn owl following the Jeffress model. In essence, we have used multiple split-gates with different widths of the ungated region on a single MoS$_2$ channel connected to the source and drain terminals to realize the computational map. Each split-gate is connected to a delay line resistor with the resistance value designed in accordance with the spatial location of the corresponding split-gate. The hallmark of our biomimetic audiomorphic computing is that digital and analog computations are performed concurrently using a single device. Note that our choice of MoS$_2$ as the semiconducting channel material is motivated by the increasing interest in two-dimensional (2D) layered materials as the successor for the aging Si technology[18–20]. The ultra-thin body nature of these 2D semiconductors not only allow for aggressive channel length scaling but also enable transparent and flexible electronics desirable for emerging technologies such as the Internet of Things (IoT)[21–24]. Furthermore, recent years have witnessed excellent progress in neuromorphic computing based on MoS$_2$ devices[25–32]. However, the concept of biomimetic audiomorphic computing need not be restricted to MoS$_2$ FETs. In fact, it can be implemented using any semiconducting material, which allows for the implementation of the split-gate FET geometry.

**Artificial coincidence detector neuron**. It is relatively straightforward to recognize that a coincident neuron acts like a two input AND gate because it fires maximum number of output spikes only when it receives signals from both the left and the

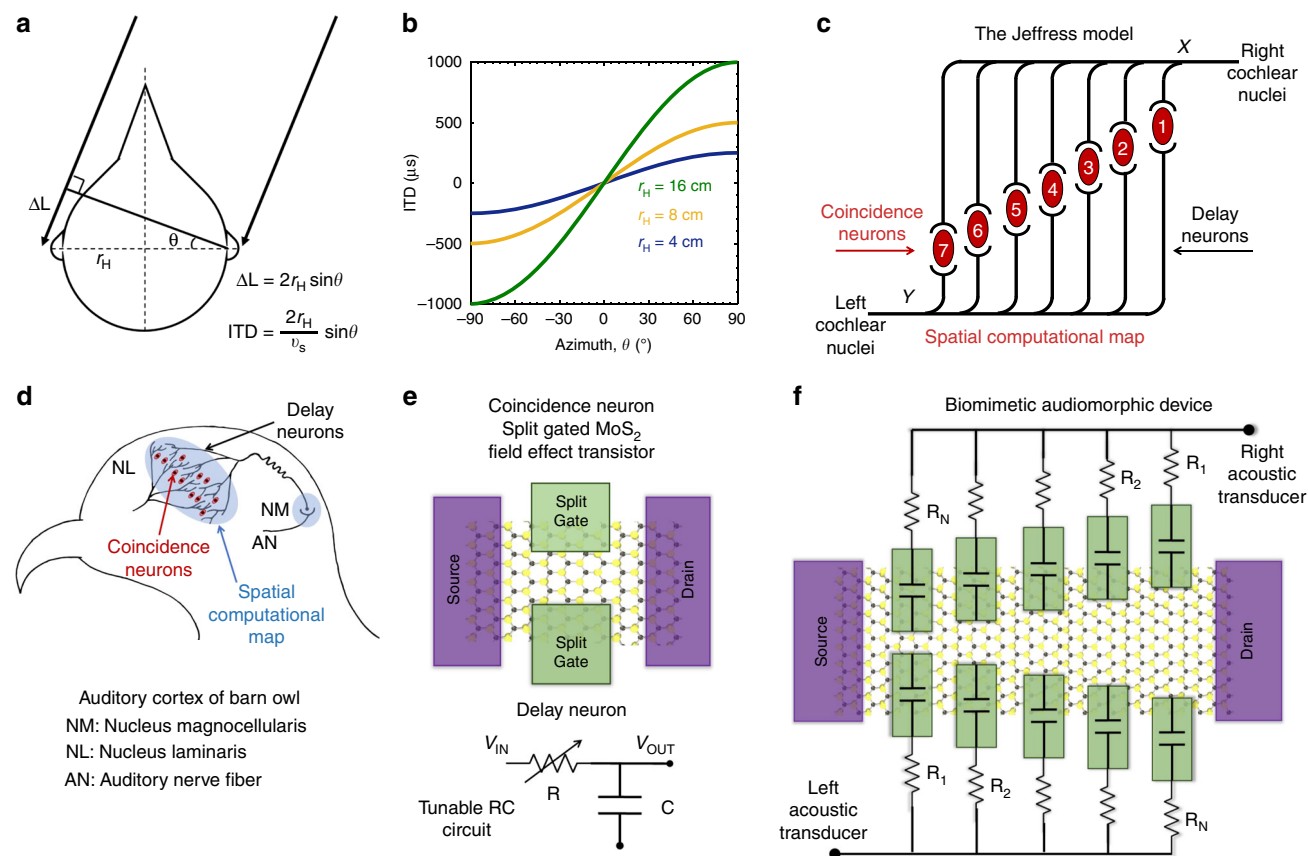

**Fig. 1** Biomimicry of auditory cortex of barn owl. **a** Path difference between the sound waves reaching the two ears and the corresponding interaural time difference (ITD) for an acoustic source located at an azimuth, $\theta$, for a given head radius ($r_H$) and sound velocity ($v_s$). **b** ITD as a function of $\theta$ for different head sizes. For typical size of animal heads, ITDs are in the range of hundreds of µs, while neuronal firing occurs only once per few ms which makes the auditory information processing a challenging task for any animal. **c** The Jeffress model showing the neural computational map in the auditory cortex for transforming temporal coding into spatial coding by deploying two key neural elements—the time delay neurons and the coincidence detector neurons. The coincidence neurons fire only when spikes arrive concurrently at the corresponding delay neurons. The delay neurons are projected onto coincidence neurons in a ladder-like branching structure. If a sound originates from straight ahead, it will reach the right and left cochlea at the same time. In this case only neuron 4 will receive coincident inputs. However, if the sound originates from the left hemisphere, it will reach the left ear earlier and coincidence will occur if the input signal from the left travels a longer path length that exactly offsets the acoustic delay, e.g., at cell 1. Similarly, if the sound originates from the right hemisphere, coincidence will occur e.g., at cell 7. The spatial projection of the delay neurons onto the coincidence neurons form a computational map where each coincidence neuron represents a specific ITD corresponding to an azimuth. **d** An anatomical drawing of barn owl's brainstem showing remarkable similarity with the Jeffress model. Axons from the nucleus magnocellularis (NM) provide the delay lines, whereas, binaural tertiary nerve fibers serve as coincidence detectors in the nucleus laminaris (NL). **e** Schematic of solid state coincidence neuron and delay neuron. Split-gated MoS₂ field effect transistors (FETs) are used to achieve the functionality of coincidence neurons and tunable RC circuits are used as delay neurons. **f** A fully integrated biomimetic audiomorphic device emulating the neural computational map involving multiple split-gates with different widths of the ungated region on a single MoS₂ channel. Each split-gate is connected to a delay line resistor with the resistance value designed in accordance with the spatial location of the corresponding split-gate

right delay neurons at the same time. It is worthwhile to mention here that coincidence circuits are known for a long time in physics as they greatly minimize the chance of a false detection. If the probability of falsely identifying a noise pulse as a genuine signal by one detector is P, then the probability of false detection when two detectors detect the signal pulse simultaneously is $P^2$. Therefore, if $P = 0.1$, then $P^2 = 0.01$. Thus, the probability of false detection can be significantly reduced by the use of coincidence detection. Figure 2a shows the schematic of a fully top-gated MoS₂ FET on a conventional Si substrate. The semiconducting MoS₂ channel is few atomic layers thick and is connected to Ni/Au metal contacts that serve as the source/drain terminals. One hundred and twenty nanometer of hydrogen silsesquioxane (HSQ) is used as the top-gate dielectric and Ni/Au is used as the top-gate electrode. The device fabrication details can be found in the experimental method section, as well as elsewhere in the literature[33]. Fig. 2b shows the transfer characteristics of the device

i.e., source to drain current ($I_{DS}$) versus top-gate voltage ($V_{TG}$) for a drain bias, $V_{DS} = 1$ V. The device is normally ON at $V_{TG} = 0$ V due to unintentional n-type doping and metal Fermi level pinning[34] close to the conduction band of MoS₂ and can be gradually turned OFF by applying negative $V_{TG}$. A high current ON/OFF ratio of ~10⁶ is achieved and the device can be treated as a one-input-one-output digital element represented by the truth table shown in Fig. 2c. Evidently this device cannot represent an AND logic. On the contrary, Fig. 2d shows the schematic of a MoS₂ FET with two split-gates separated by 200 nm. Figure 2e shows the transfer characteristics of the device i.e., source to drain current ($I_{DS}$) versus split-gate voltage ($V_{SG}$) for a drain bias, $V_{DS} = 1$ V under two different conditions. The red curve depicts the device characteristics when one of the split-gates is swept from 0 V to −30 V, while the other split-gate is held at a constant bias of 0 V, whereas, the blue curve depicts the device characteristics when both split-gates are simultaneously swept

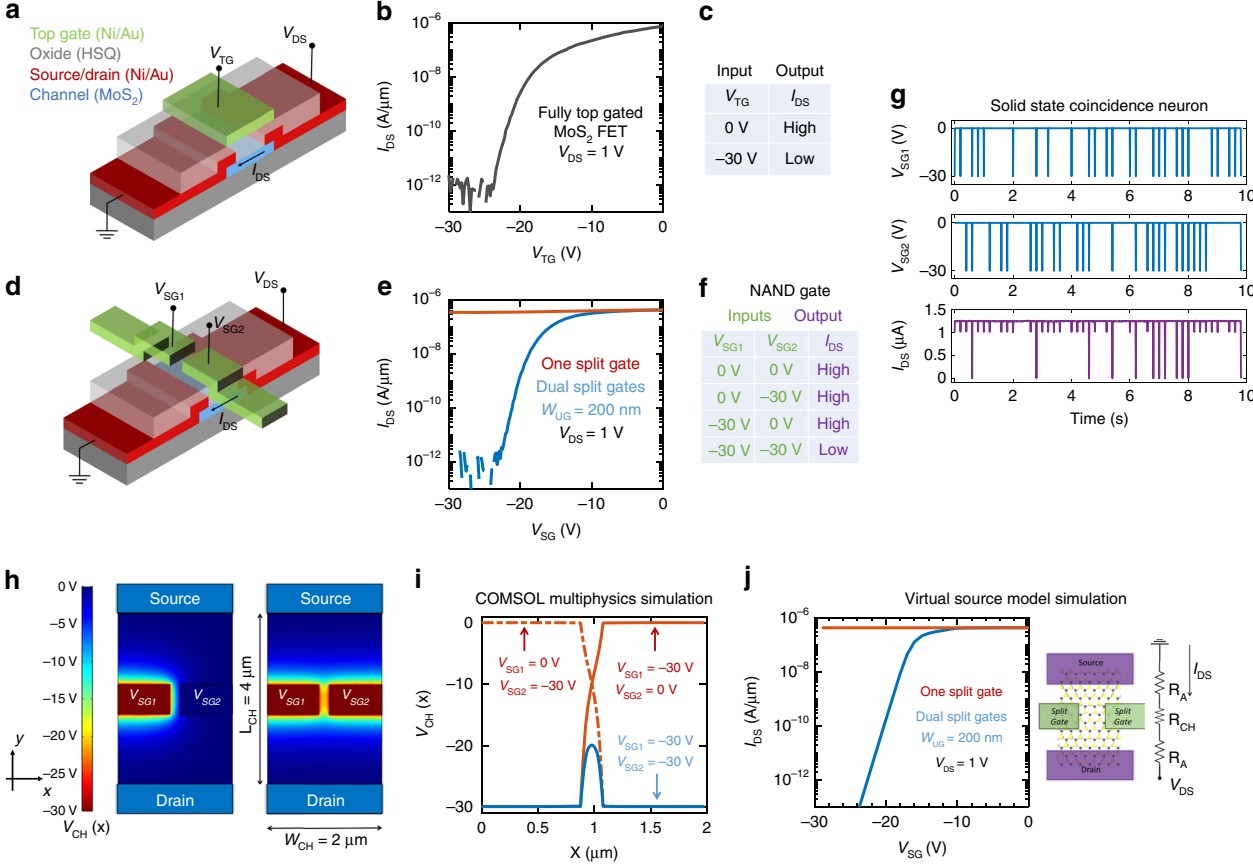

**Fig. 2 Artificial coincidence detector neuron. a** Schematic and (**b**) Transfer characteristics of a fully top-gated MoS$_2$ field effect transistor (FET) with 120 nm of hydrogen silsesquioxane (HSQ) as the top-gate dielectric and Ni/Au as the top-gate electrode. MoS$_2$ channel is few nm thick and is connected to Ni/Au metal contacts that serve as the source/drain terminals. The device is normally ON at $V_{TG} = 0$ V and can be switched OFF by applying $V_{TG} = -30$ V with a high current ON/OFF ratio of ~10$^6$. **c** Truth table showing that the device can be regarded as a one-input-one-output digital element. **d** Schematic of an MoS$_2$ FET with two split-gates separated by an ungated region of width $W_{UG} = 200$ nm. **e** Transfer characteristics of the split-gated device when one of the split-gates is swept from 0 V to −30 V while the other split-gate is held at a constant bias of 0 V (red curve) and when both split-gates are simultaneously swept from 0 V to −30 V (blue curve). **f** Truth table showing that the split-gated device can be treated as two-input-one-output digital element with NAND logic. **g** Random sequence of voltage pulses of magnitude −30 V are applied to the two spilt gates, $V_{SG1}$ and $V_{SG2}$. The output current is completely suppressed or inhibited only when the spikes coincide suggesting that the split-gated MoS$_2$ FET can be used to mimic neural coincidence. **h** COMSOL multiphysics simulation of the 2D potential profile when −30 V bias is applied to either one or both split-gates. **i** 1D potential profile along the channel width for different combinations of the two split-gate biases shows the effect of fringing electric field and capacitive coupling between the two split-gate electrodes. The channel potential in the ungated region between the split-gates is finite under all conditions. The effect is more dramatic when $V_{SG1} = V_{SG2} = -30$ V. **j** Simulated transfer characteristics of the split-gated MoS$_2$ FET using the Virtual Source (VS) model and the electrostatic potential profile, $V_{CH}(x)$ along the channel width obtained from the COMSOL simulations. We have used a modified VS model to calculate channel resistance, $R_{CH}$ that captures the variation in the electrostatic potential along the width of the channel and also to account for the access resistance, $R_A$ due to the ungated region along the channel length

from 0 V to −30 V. Unlike the fully top-gated case, $I_{DS}$ is relatively high when one of the split-gate is at −30 V. In other words the device cannot be turned OFF by one split-gate and can be considered as always ON. However, when both split-gates reach −30 V simultaneously, the device reaches OFF state with current ON/OFF ratio of ~10$^6$. Furthermore, the device can be considered as a two-input-one-output digital element represented by the truth table shown in Fig. 2f, which can be identified as the NAND logic. Figure 2g shows that the output current from the device is completely suppressed only when $V_{SG1}$ and $V_{SG2}$, of magnitude −30 V, arrive at the two corresponding spilt gates, concurrently. The width of the voltage spikes was 10 ms. Figure 2g confirms that a split-gated MoS$_2$ FET can be used as a coincidence detector neuron. Figure 2h shows the COMSOL multiphysics simulation of the 2D potential profile when −30 V bias is applied to either one or both split-gates. Figure 2i shows the 1D potential profile

along the channel width for different combinations of the two split-gate biases. Note that due to the fringing electric field and capacitive coupling between the two split-gate electrodes the potential does not abruptly drop to zero at the edge of the split-gates when $V_{SG1} = -30$ V and $V_{SG2} = 0$ V and vice versa (solid and dotted red lines). This effect is even more dramatic when $V_{SG1} = V_{SG2} = -30$ V. Under this scenario, the channel potential in the ungated region is far from being zero or in other words the ungated region becomes partially gated by the split-gates (solid blue line). Figure 2j shows the simulated transfer characteristics of the split-gated MoS$_2$ FET using the Virtual Source (VS) model[35,36] (see Supplementary Note 1 for details) and the electrostatic potential profile, $V_{CH}(x)$ along the channel width obtained from the COMSOL simulations. Clearly, the simulated split-gated device characteristics shown in Fig. 2j reflects the experimentally measured device data shown in Fig. 2e.

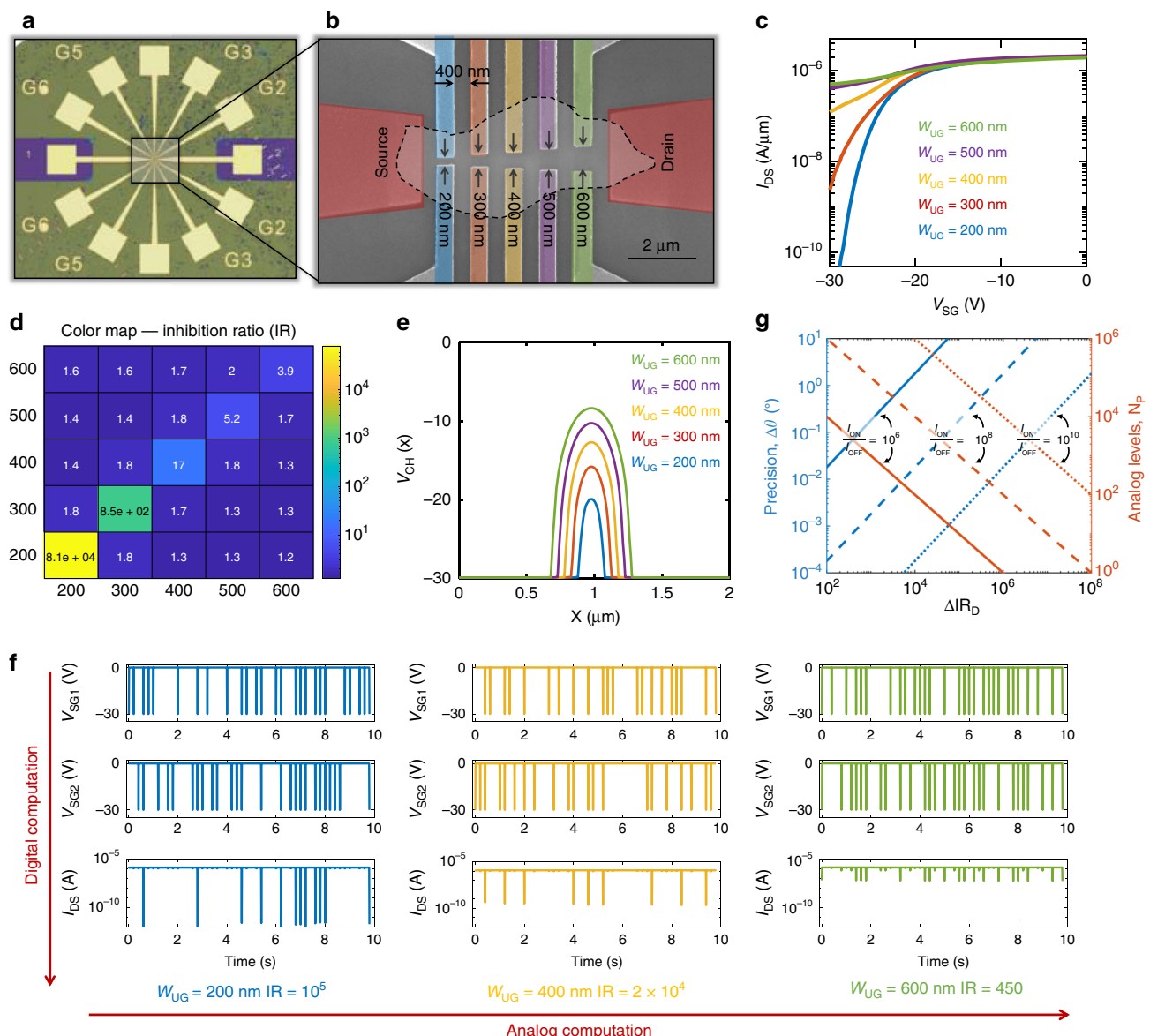

**Fig. 3** Construction of neural auditory computational map. **a** Optical image and (**b**) zoomed in scanning electron microscope (SEM) image (false colored) of a biomimetic device for imitating the neural computational map inside the auditory cortex of barn owl. The device consists of multiple split-gates distributed along the length of the semiconducting MoS$_2$ channel which is connected to the source and the drain metal contacts. These split-gates have constant lateral spacing of 400 nm and monotonically increasing vertical spacing from 200 nm to 600 nm. **c** Transfer characteristics of the device when each of the five colored pairs of split-gates are concurrently swept from 0 V to −30 V. **d** Color map of the inhibition ratio (IR) for all possible combinations of the split-gate pairs. IR is defined as the ratio of current through the device corresponding to $V_{SG} = 0$ V and $V_{SG} = −30$ V, concurrently applied to a given split-gate pair. IRs are significantly larger for split-gate pairs that are vertically aligned (diagonal elements in the color map). Within these vertically aligned pairs, the one with minimum split-gate spacing has the maximum IR and vice versa. **e** COMSOL simulation showing the 1D potential profile across the channel width for different split-gate spacing for $V_{SG} = −30$ V applied to both split-gates. The magnitude of the potential in the ungated segment is much closer to the split-gate potential when the spacing is 200 nm compared to 600 nm and follows a monotonic trend as a function of the split-gate spacing. **f** Output current from the device when random voltage spikes of magnitude −30 V are applied to the two spilt gates corresponding to different spatial pairs. All pairs detect coincidence by suppressing the device current which is a digital operation, whereas, the device as a whole determines the location of coincidence through analog IR values. **g** Angular precision (Δθ) and corresponding number of analog computation levels ($N_P$) as a function of the minimum difference between the analog IR values corresponding to two consecutive split-gate pairs (ΔIR$_D$) for different current ON/OFF ratio of the semiconducting channel. Clearly, the biomimetic audiomorphic device can be designed to offer orders of magnitude better precision than the barn owl

**Construction of neural auditory computational map.** In order to mimic the computational map constructed by the coincidence neurons in the auditory cortex of the barn owl, we fabricated a device structure as shown using the optical image in Fig. 3a and a false colored scanning electron microscope image in Fig. 3b. The device consists of multiple split-gates distributed along the length of the MoS$_2$ channel. These split-gates have constant lateral

spacing of 400 nm and monotonically increasing vertical spacing from 200 nm to 600 nm. Figure 3c shows the transfer characteristics of the device when each of the five colored pairs of split-gates are concurrently swept from 0 V to −30 V. Note that a split-gate pair can, in principle, be constructed by combining any one of the colored gates from the top with any other one from the bottom. In fact there exists N(N + 1)/2 distinct pairs of split-gates

in total, which in our case is 15 since $N = 5$. Here, we introduce a parameter called inhibition ratio (IR) which is defined as follows:

$$\text{IR} = \frac{I_{\text{DS}}(V_{\text{SG}} = 0\,\text{V})}{I_{\text{DS}}(V_{\text{SG}} = -30\,\text{V})} \quad (1)$$

IR is the ratio of current through the device corresponding to $V_{\text{SG}} = 0\,\text{V}$ and $V_{\text{SG}} = -30\,\text{V}$, concurrently applied to any given split-gate pair. Figure 3d shows the IR color map for all possible combinations of the split-gate pairs. Noticeably the IRs are significantly larger for split-gate pairs that are vertically aligned (diagonal elements in the color map). These are the ones which will be used as individual coincidence detector neurons. Furthermore, within these vertically aligned pairs, the one with minimum split-gate spacing has the maximum IR and vice versa. Figure 3e shows the COMSOL simulation results for the 1D potential profile across the channel width for different split-gate spacing for $V_{\text{SG}} = -30\,\text{V}$ applied to both split-gates. As expected the magnitude of the potential in the ungated segment is much closer to the split-gate potential when the spacing is 200 nm compared to 600 nm, and follows a monotonic trend as a function of the split-gate spacing. From the color map it is clear that the IR drops almost exponentially as a function of the split-gate spacing. This is interesting since the IR is mostly determined by the current that flows through the ungated region at $V_{\text{SG}} = -30\,\text{V}$. Note that at this $V_{\text{SG}}$ the gated regions are switched OFF. Although the electrostatic potential of the ungated region due to the split-gate potential diminishes linearly with the split-gate spacing, the exponential dependence in IR can be explained from the fact that the device is biased close to the subthreshold regime, where the device current is an exponential function of the channel potential. Note that the IR color map can be used to identify the split-gate pair, where the coincidence took place or in other words the device constructs a spatial map for the coincidence detection. Figure 3f shows the output current from the device when random voltage spikes of magnitude $-30\,\text{V}$ are applied to the two split gates corresponding to different spatial pairs. All pairs detect coincidence by suppressing the device current. Furthermore, the suppression is maximum with an IR of $\sim 10^5$ when coincidence occurred in the split-gate pair corresponding to 200 nm vertical spacing and minimum with an IR of $\sim 450$ for 600 nm vertical spacing. Clearly, individual split-gate pairs perform digital computation for coincidence detection (NAND logic), while, the device as a whole uses analog IR values for determining the spatial location of the coincidence. As such our biomimetic audiomorphic device is a new precedence, where a single device seamlessly combines digital and analog computation—a feature that is abundant in biological neural networks. It should be further noted that the number of split-gate pairs or coincidence neurons, $N_{\text{P}}$, determines the number of analog computation levels which in turn determines the angular precision, $\Delta\theta$, of the biomimetic audiomorphic device following the relationship, $\Delta\theta = 180^0/N_{\text{P}}$. From this expression, it may appear that the angular precision can be made arbitrarily small by increasing the number of analog levels. However, a large number of analog levels would require a wider analog range for the current suppression or IR or in other word it necessitates a semiconducting channel with large current ON/OFF ratio. If the minimum required difference between the analog IR values corresponding to two consecutive split-gate pairs is $\text{IR}_{\text{D}}$, then $N_{\text{P}}$ and hence $\Delta\theta$ will be determined by the following expression.

$$N_{\text{P}} = \frac{I_{\text{ON}}/I_{\text{OFF}}}{\Delta\text{IR}_{\text{D}}}; \Delta\theta = 180^0\,\frac{\Delta\text{IR}_{\text{D}}}{I_{\text{ON}}/I_{\text{OFF}}} \quad (2)$$

Figure 3g shows $\Delta\theta$ and $N_{\text{p}}$ as a function of $\Delta\text{IR}_{\text{D}}$ for different values of $I_{\text{ON}}/I_{\text{OFF}}$. Note that a small value for $\Delta\text{IR}_{\text{D}}$ results in greater precision but also requires larger number of split-gate pairs. Furthermore, small $\Delta\text{IR}_{\text{D}}$ can be more susceptible to noise. Interestingly, the same precision can be achieved for a larger $\Delta\text{IR}_{\text{D}}$ by increasing the ON/OFF ratio. Nevertheless, the biomimetic audiomorphic device can be designed to offer orders of magnitude better precision than the barn owl. The biomimetic device also offers advantages in device footprint and scalability as discussed in the Supplementary Note 2 and Supplementary Note 3.

**Artificial time delay neurons.** As described in the Jeffress model, the auditory cortex exploits the finite axonal conduction velocity for transforming the binaural acoustic information encoded via ITDs into a spatial computational map for sound localization. However, unlike the solid state electronic circuits where the signal propagation through metallic interconnects occur at a very high speed, axonal signal propagation is significantly slower. This is because the propagation of action potential along the length of an axon requires periodic charging of the leaky axoplasomic membrane. In vertebrate axons, the myelin sheath improves the axonal insulation and action potentials are regenerated only at the nodes of Ranvier through high density sodium ion-channels[37]. Fig. 4a shows the basic neuro anatomy of a vertebrate axon and its equivalent circuit representation following the distributed transmission line model. Classical cable theory can be used to describe the signal propagation through the axon by involving two critical parameters, the length constant ($\lambda$), which determines how far the action potential can propagate without regeneration, and, the time constant ($\tau$), which determines how fast the action potential can propagate, resulting in a finite axonal conduction velocity $v_{\text{A}} = \lambda/\tau$. Intracellular recordings from the NM axons in NL reveal a conduction velocity of 3–5 m s$^{-1}$[17]. More discussion on axonal conduction velocity can be found in Supplementary Note 4. Figure 4b shows a 3D plot for the length of delay line ($L_{\text{D}}$) required for mapping the ITDs ($t_{\text{ITD}}$) as a function of the signal propagation velocity ($v_{\text{p}}$) following the expression: $L_{\text{D}} = v_{\text{p}}t_{\text{ITD}}$. The red circle indicates that $\sim 1$ m long metallic interconnect is necessary to map ITDs in the range of 100 µs. This is impractical for any integrated circuit and arises from the fact that the speed of electronic signal transmission through metal is relatively fast at $\sim 10^6$ m s$^{-1}$. On the contrary, as indicated by the green circle, the barn owl accomplishes the same by using an intermodal distance of only $\sim 60$ µm[17] for the axons that act as delay lines within the NL. This is simply due to the orders of magnitude slower signal propagation through the axons as discussed earlier. Therefore, it is necessary to introduce solid-state delay elements into our biomimetic device in order to capture the time scale of axonal conduction and thereby reduce the required wire length. Figure 4c shows the schematic of an artificial time delay neuron realized by connecting the gate terminal of one MoS$_2$ FET to the drain terminals of another MoS$_2$ FET. In principle, we have used the gate capacitance ($C_{\text{G}}$) and channel resistance ($R_{\text{CH}}$) of the corresponding MoS$_2$ FETs to construct a simple RC circuit. Figure 4d shows the experimental transient responses of our artificial time delay neuron which can be captured using Eq. 3:

$$V_{\text{OUT}} = V_{\text{IN}}\left[1 - \exp\left(-\frac{t}{R_{\text{CH}}C_{\text{G}}}\right)\right] = V_{\text{IN}}\left[1 - \exp\left(-\frac{t}{\tau_{\text{C}}}\right)\right] \quad (3)$$

In Fig. 4d the different transient responses correspond to different time constants or delays ($\tau_{\text{C}}$) achieved by biasing the

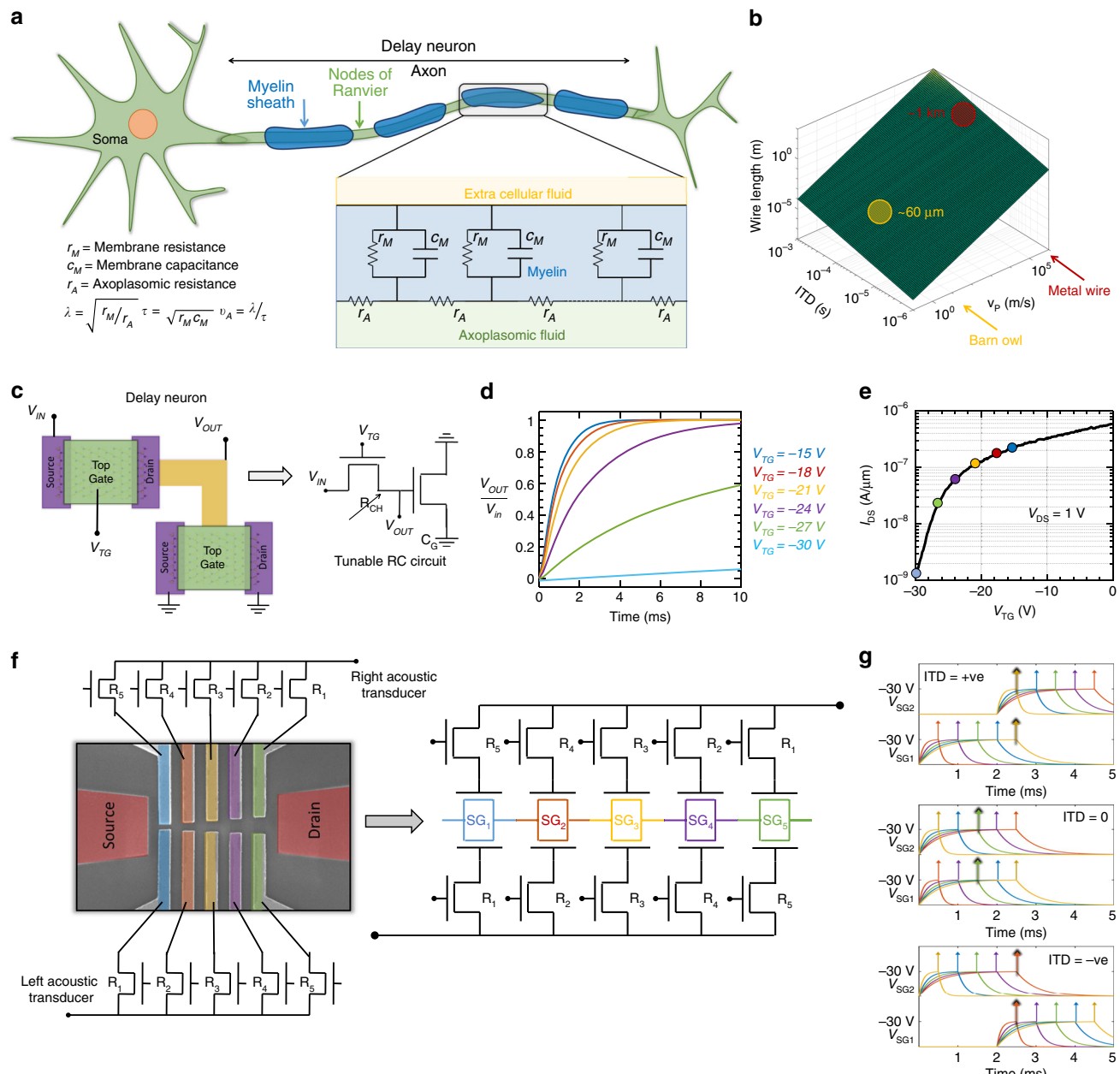

**Fig. 4 Artificial time delay neuron. a** Schematic of a vertebrate axon and its equivalent circuit representation following the distributed transmission line model. The length constant ($\lambda$) determines how far the action potential can propagate without regeneration, and, the time constant ($\tau$) determines how fast the action potential can propagate, resulting in a finite axonal conduction velocity $v_A = \lambda/\tau$. **b** Length of delay line required for mapping the ITDs as a function of the signal propagation velocity ($v_p$). The axonal conduction velocity in barn owl is 3–5 m/s translating into few tens of $\mu$m axonal length required for mapping the ITDs, whereas metallic interconnects will require impractical wire lengths necessitating the introduction of delay elements into biomimetic device. **c** Schematic of an artificial delay neuron realized by connecting the gate terminal of one MoS$_2$ FET to the drain terminals of another MoS$_2$ FET. The gate capacitance ($C_G$) and channel resistance ($R_{CH}$) of the corresponding MoS$_2$ FETs construct a tunable RC circuit. **d** Experimental transient responses of solid state delay neuron with tunable time constants or delays ($\tau_c$) achieved through different top-gate biases applied to the resistive MoS$_2$ FET. **e** Transfer characteristics of the resistive MoS$_2$ FET at $V_{DS} = 1$ V. **f** Fully integrated biomimetic audiomorphic architecture. The resistances of the MoS$_2$ FETs connected to the bottom halves of the split-gates increase monotonically from left to right and vice versa for the top halves of the split-gates. **g** Each split-gate is charged in response to audio cues. The spikes mark the times when the respective gates reach their peak negative potential. Color coding shows that the coincidence occurs at the central split-gate (green) for ITD = 0, and on the right side (yellow) for positive ITD and left side (red) for negative ITD

resistive MoS$_2$ FET in different regimes of operations as shown in Fig. 4e. The time constant was found to vary from ~800 $\mu$s at $V_{TG} = -15$ V to ~200 ms at $V_{TG} = -30$ V corresponding to $R_{CH}$ values varying from 4.4 M$\Omega$ to 0.9 G$\Omega$, respectively. The gate capacitance due to large gate metal pads and other parasitic contributions was 180 ± 20 pF. The time constant can be reduced

by reducing the $R_{CH}$ by biasing the fully top-gated MoS$_2$ FET further into the ON-state. Finally, Fig. 4f shows the complete biomimetic audiomorphic architecture where each split-gate is connected to the drain terminal of a corresponding fully top-gated MoS$_2$ FET. Note that the resistances of the MoS$_2$ FETs connected to the top and bottom split-gates increase

monotonically from right to left and left to right, respectively, imitating the organization of delay axons in the auditory cortex of barn owl. The required resistance values can be achieved by designing the length and width of the $MoS_2$ FETs accordingly. Figure 4g shows how each split-gate is charged when the acoustic transducers at the two ends of the chip receive the sound signal with positive, zero and negative interaural delays (ITDs). The spikes mark the times when the respective gates reach their peak negative potential. Color coding shows that the coincidence occurs at the central split-gate (green) for ITD = 0, and on the right side (yellow) for positive ITD and left side (red) for negative ITD.

**Neuroplasticity**. One of the remarkable features of brain is its plasticity[38]. For example barn owls survive by hunting small rodents in the open countryside in Europe. However, there is a significant change in the ground cover available to its prey species between winter and summer[39,40]. The barn owl must, therefore, become more precise and alert in sound localization in winter than in summer by reversibly changing its brain morphology. Brain morphology can also change due to adaptation to local environment[41]. For example, the precision in sound localization differs between the barn owls found in the Mediterranean versus Europe or North America. Therefore, it is important to introduce neuroplasticity in our solid-state computing devices. Figure 5a shows the schematic of the device with global back-gating capability. The back-gate dielectric is 285 nm $SiO_2$ and the back-gate electrode is heavily doped Si. Figure 5b shows the back-gate transfer characteristics for $V_{DS} = 1$ V. The back-gate has full electrostatic control over the entire channel, which is reflected in the large current ON/OFF ratio of $10^6$. The back-gate threshold voltage was found to be $V_{TB} = -12$ V. Figure 5c shows the transfer characteristics of the biomimetic device when the split-gate pair with 200 nm spacing are concurrently swept from 0 V to $-30$ V under different back-gate ($V_{BG}$) biases. Clearly, the inhibition ratio (IR) changes significantly and non-monotonically as a function of $V_{BG}$ as shown in Fig. 5d (blue dotted lines and circles). This can be explained from the fact that when, $V_{BG} \gg V_{TB}$, the device is in the deep ON-state, and hence the current through the ungated region remains high even for $V_{SG} = -30$ V resulting in low IR. As $V_{BG}$ approaches $V_{TB}$, but still remains above threshold, the current through the device for $V_{SG} = 0$ V only reduces linearly. However, now it becomes relatively easier for the split-gates to switch the ungated channel region from ON state to subthreshold and reduce the device current exponentially for $V_{SG} = -30$ V resulting in an increased IR. Once $V_{BG}$ goes below $V_{TB}$, the entire device enters subthreshold operation. Now the current through the device for $V_{SG} = 0$ V also reduces exponentially resulting in a decrease in IR. Finally, for $V_{BG} \ll V_{TB}$, the entire device is in the deep subthreshold regime with current levels beyond the detection limits of the measurement apparatus, and as such the effect of $V_{SG}$ becomes inconsequential. Figure 5d also shows the IR as a function of $V_{BG}$ for different split-gate pairs. The non-monotonic trend is observed for all split-gate pairs, however, the $V_{BG}$ value for which a given split-gate pair achieves maximum inhibition ratio decreases monotonically as the split-gate spacing increases. This is expected since larger split-gate spacing translates into weaker split-gate control of the ungated region, and hence it requires significant electrostatic aid from the back-gate. In other words the back-gate bias should be close to the back-gate threshold so that the impact of the split-gate pair is stronger in the ungated region. In a dual-gated FET, the coupling between the top/split-gate and the back-gate is critical in understanding the impact of one on the other. Figure 5e shows the back-gate transfer characteristics of a fully top-gated

$MoS_2$ FET for different top-gate voltages ($V_{TG}$) at $V_{DS} = 1$ V. The top-gate transfer characteristics of the same $MoS_2$ FET for different back-gate voltages ($V_{BG}$) is shown in the Supplementary Fig. 1. Clearly, the back-gate threshold voltage, $V_{TB}$, depends on $V_{TG}$, which can be explained from the principle of charge balance, i.e., the inversion charge induced by the top-gate must be compensated by the back-gate and vice versa. Figure 5f shows the iso-current $V_{TB}$ extracted from Fig. 5e as a function of $V_{TG}$. Note that the slope of Fig. 5f is proportional to the ratio of top-gate capacitance to back-gate capacitance, i.e., $C_{TG}/C_{BG}$, which was found to be ~1.7, consistent with the thicknesses of ~120 nm and ~285 nm and dielectric constants of ~3.2 and ~3.9 of HSQ and $SiO_2$, respectively. These results were used in conjunction with the COMSOL simulations in the VS model to obtain the split-gate transfer characteristics for different $V_{BG}$ values as shown in Fig. 5g for a split-gate spacing of 200 nm. Figure 5h shows the extracted IR from the VS model simulation results as a function of $V_{BG}$ for various split-gate spacing ranging from 50 nm to 600 nm. To mimic the experimental conditions, we restricted the minimum OFF current through the device to 50 pA/μm corresponding to the leakage floor of the measurement instrument. Clearly, the simulated split-gate transfer characteristics and the IR plots show remarkable similarity with the experimental results. Therefore, we can conclude that the back-gate bias can be used to tune the IR by orders of magnitude and hence can be used to imitate neuroplasticity. Figure 5i shows the color map of IR for all possible split-gate pairs of our biomimetic device under different back-gate biases. Note that a large differences in the magnitude of IR among the diagonal elements is desirable for easier detection of the location of coincidence and hence the localization of the sound source. Clearly, $V_{BG} = 14$ V provides the maximum IR contrast and hence we can relate the corresponding color map to the hyper-attentive state of the barn owl. As expected, the IR contrasts in the color maps diminish monotonically as $V_{BG}$ becomes more positive i.e., the device is biased in the deep ON-state. Therefore, the IR color maps corresponding to $V_{BG} = 18$ V, 22 V, and 26 V can be correlated to the attentive, wakeful and resting state of the barn owl, respectively.

**Discussion**

In the conclusion, we have successfully demonstrated the biomimicry of the neural computational algorithm in the auditory cortex of barn owl. We have used a device that consists of multiple split-gates on a single semiconducting $MoS_2$ channel connected to source/drain contacts for emulating the spatial map of coincidence detector neurons and tunable RC circuits for imitating the time delay neurons following the Jeffress model of sound localization. The unique feature of this biomimetic audiomorphic device is the fact that it seamlessly combines digital and analog computation, which is abundant in the biological neural networks. In short, individual split-gate pairs perform digital computation using NAND logic for determining spiking coincidence, while, the device as a whole uses analog values for the inhibition ratio (IR) to determine the spatial location of the coincidence. Further, artificial time delay neurons realized using the semiconducting channel resistance and gate capacitance of $MoS_2$ FETs allow biomimicking of finite axonal conduction velocity, which is essential for translating the ITDs into a spatial computational map. Finally, global back-gating capability adds tunable and reversible neuroplasticity to the biomimetic audiomorphic device. While the Jeffress model is applicable for many animals that use the ITDs to locate the source of low frequency sounds, recent research has also shown a wider range of neural computational algorithms in the auditory cortex of avian, mammalian, and reptilian nervous systems, which will be the topic of our future research.

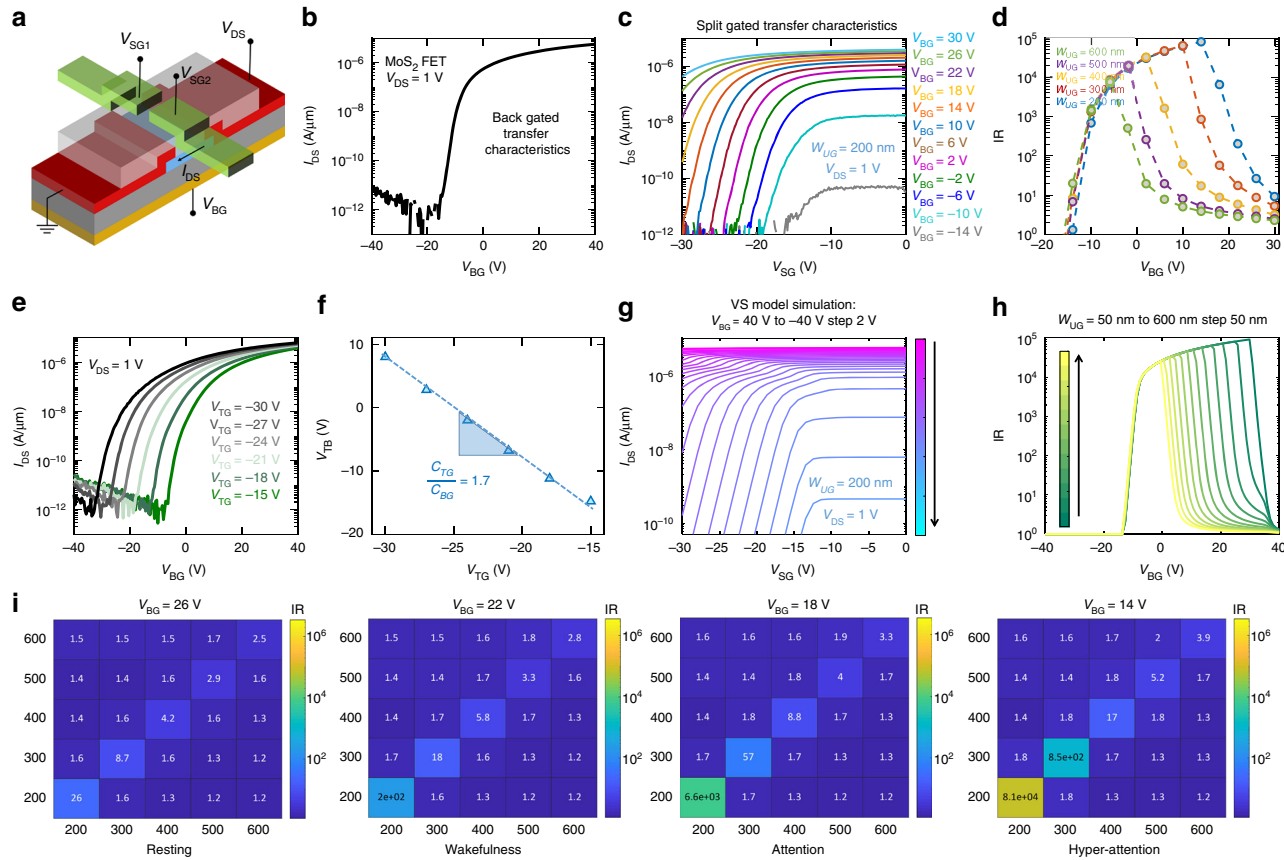

**Fig. 5** Neuroplasticity in biomimetic audiomorphic device. **a** Schematic of the biomimetic device with global back-gate using 285 nm SiO$_2$ as the dielectric and heavily doped Si as the back-gate electrode. **b** Back-gate transfer characteristics for $V_{DS} = 1$ V. The back-gate threshold voltage was $V_{TB} = -12$ V. **c** Transfer characteristics of the biomimetic device when the split-gate pair with 200 nm spacing are concurrently swept from 0 V to $-30$ V under different back-gate ($V_{BG}$) biases. **d** Inhibition ratio (IR) as a function of $V_{BG}$ for different split-gate spacing extracted from the corresponding transfer characteristics shows a non-monotonic trend. For, $V_{BG} \gg V_{TB}$, the device is in the deep ON state and hence the current through the ungated region remains high for $V_{SG} = -30$ V resulting in low IR. As $V_{BG}$ approaches $V_{TB}$, the current through the device for $V_{SG} = 0$ V only reduces linearly but since it becomes relatively easier for the split-gates to switch the ungated channel region from ON state to subthreshold for $V_{SG} = -30$ V reducing the device current exponentially and resulting in an increased IR. Once $V_{BG}$ goes below $V_{TB}$, the entire device enters subthreshold operation. Now the current through the device for $V_{SG} = 0$ V also reduces exponentially resulting in a decrease in IR. **e** Back-gated transfer characteristics of a fully top gated MoS$_2$ FET for different top-gate voltages ($V_{TG}$) at $V_{DS} = 1$ V. **f** Iso-current $V_{TB}$ as a function of $V_{TG}$ shows the strength of coupling between two gated in a dual gated geometry. **g** Simulated transfer characteristics of split-gated device with 200 nm spacing using the VS model, COMSOL simulation and experimentally extracted coupling between the top/split-gate and the back-gate. **h** IR extracted from the simulation results as a function of $V_{BG}$ for various split-gate spacing ranging from 50 nm to 600 nm. The simulated characteristics show remarkable similarity with the experimental results suggesting that the back-gate bias can be used to tune the IR by orders of magnitude and hence can be used to imitate neural plasticity. **i** IR color maps for all possible split-gate pairs of the biomimetic device under four different back-gate biases. $V_{BG} = 14$ V provides the maximum IR contrast and can be related to the hyper-attentive state of the barn owl, whereas, diminishing IR contrasts with increasing $V_{BG} = 18$ V, 22 V, and 26 V can be correlated to the attentive, wakeful and resting state of the barn owl, respectively

## Methods

**Device fabrication and measurements**. The top-gated and split-gated devices were fabricated using micromechanically exfoliated MoS$_2$ flakes on 285 nm thermally grown SiO$_2$ substrates with highly doped Si as the back-gate electrode. The source/drain contacts were defined using electron-beam lithography (Vistec EBPG5200). Ni (40 nm) followed by Au (30 nm) were deposited using electron-beam evaporation for the contacts. For fabricating the top-gate, hydrogen silsesquioxane (HSQ) was used as the dielectric. It was deposited by spin coating 6% HSQ in methyl isobutyl ketone (MIBK) (Dow Corning XR-1541-006) at 4000 rpm for 45 s and baked at 80 °C for 4 min. The HSQ was patterned using an e-beam dose of 2000 μC/cm$^2$ and was developed at room temperature using 25% tetramethylammonium hydroxide (TMAH) for 30 s following a 90 s rinse in deionized water (DI). Next, it was cured in air at 180 °C and then 250 °C for 2 min and 3 min, respectively. The top-gate electrode was patterned using electron-beam lithography followed by the deposition of Ni/Au using electron-beam evaporation as the contact. The electrical characterization was carried out at room temperature in high vacuum (~10$^{-6}$ Torr) in a Lake Shore CRX-VF probe station using a Keysight B1500A parameter analyzer.

## Data availability

The datasets generated during and/or analyzed during the current study are available from the corresponding author on reasonable request.

## Code availability

The codes used for simulation data are available from the corresponding authors on reasonable request.

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

## Acknowledgements

The work was partially supported through Grant Number FA9550-17-1-0018 from Air Force Office of Scientific Research (AFOSR) through the Young Investigator Program.

## Author contributions

Saptarshi Das conceived the idea, designed the experiments, and wrote the paper. Sarbashis Das, Akhil Dodda, and Saptarshi Das performed the experiments and analyzed the data. All the authors discussed the results, agreed on their implications, and contributed to the preparation of the paper.

## Additional information

**Competing interests:** The authors declare no competing interests.

