## [Peer Review File · Nature Communications]

Reviewers' comments:

Reviewer #1 (Remarks to the Author):

The manuscript by Das et al. described a multi-gate MoS₂ FET. Each pair of the gate serves as a NAND for coincidence detection. In addition, the inputs to gates see different RC with different gates, achieved by wiring extra FETs as resistors. Since paired gates are of different gaps, the multi-gate FET produces different analog currents upon different time gaps between the signals, like the Jeffress's model for the sound detection. In addition, back gate modulation is also investigated. The results are interesting and timely. However, there are some questions to be addressed.

First, multi-gate logic has long been studied. The authors need to show the advantages (e.g. footprint, scalability, performance like S/N ratio) that a multi-gate FET possesses over the functional equivalent logic formed by conventional FETs. Say for instance, a NAND of 2 single-gate FETs connected in parallel.

Second, the detection in Fig. 3 is based on analog I_{ds} currents, which demands extra components like ADCs that are not counted here. The authors may like to evaluate the system level performance and compare that with conventional logic circuits (e.g. multiple NANDs sharing common inputs. Each NAND has its dedicated output, rather than a shared common analog output.).

Third, both the abstract and the main text are relatively lengthy, with probably too much background information. The referee agrees that the background information is mandatory, but too much background may dilute the focus. The referee suggests making the writing concise, with introduction tightly related to the background.

The detailed technical comments are as following.

1. The abstract could be more concise to highlight what is the primary novelty, and what's the secondary.
2. The introduction comprises psychological claims that lack references. The referee doubts whether it is valid to claim human learning is driven by curiosity while that of animals is not. In addition, the referee tends to feel that human behaviors are essentially still demands driven, but humans could

foresee the long-term consequences of actions. In short, references are needed for claims. And as mentioned, the introduction should be more closely related to the main achievement, which is the simulation of Jeffress's model.

3. The referee wonders what are the advantages of using HSQ as a gate oxide. Could HSQ make a good interface with the MoS₂? What is the mobility difference between the FET with HSQ gate and that with back gate alone?

4. As mentioned in the general comment, the advantages of the multi-gate FET need to be pointed out. On the contrary, the logic equivalent NAND with a pair of single-gate FETs could be benefited with less gate coupling and less fringing electric field.

5. What is the temporal width of the voltage pulses in Fig. 2g? How fast could this logic gate switch?

6. The referee suggests combining Fig. 2j with the experimental curve and making discussion concise, unless the simulation results are related to the novelty of this work.

7. It will be more helpful to highlight the MoS₂ flake in Fig. 3a or b.

8. Regarding the Fig. 4c-e, is this two-transistor scheme realized on the same chip? Or these devices are wired via probes?

9. How are the capacitance (180 ± 20 pF) of the different gates measured? Why the gate capacitances do not scale with the gap distance of paired gates given the back gate is just 285 nm?

10. For Fig. 4f, what are the advantages of using FETs to cause RC delay rather than just resistors? Does the delay need to be tuned?

11. It may be better to include the original waveforms in Fig. 4g, in addition to indications of the places where peak voltages are reached.

12. Usually neuroplasticity is due to synaptic plasticity that changes the connection strengths between neurons. Is it proper to term back gate effect as an analog of the neuroplasticity?

Reviewer #2 (Remarks to the Author):

This manuscript by Das et al. demonstrates an electronic device based on a 2D semiconductor MoS₂ to mimic the spatiotemporal responses needed for sound localization. The device consists of a MoS₂ field-effect transistor with multiple top-gate electrodes. The top gates are fabricated in pairs with varying gaps such that local conductivity of the channel is controlled by simple electrostatic control in the gap. The field is topical. However, as listed below, there are several serious problems with this manuscript that make it unsuitable for Nature Communications.

- 1) Showing sound localization by spatiotemporal responses in MoS₂ transistors has been shown before in this paper Nano Lett., 2018, 18 (5), 3229. The authors do not acknowledge this paper or even cite it. So, the major claim of novelty has already been taken and ignoring that fact is not helping here.
- 2) Although the authors show a tremendous amount of data in the paper the full sound localization operation is actually not demonstrated. Physically, two different neurons are needed - coincidence neurons and time-delay neuron. The authors show the operation of coincidence neuron in much details in Fig. 1 - 3. The individual time-delay neuron responses are shown in Fig. 4d, e. But I am not seeing the integration of the two for sound localization. There is a schematic in Fig. 4f but did they actually make the circuit? If so, the experimental data should be presented.
- 3) Another related issue is why anyone would want to do sound localization with MoS₂ transistors? First of all, there are quite advanced silicon CMOS chips which are used in commercial robotics and navigation system. The authors may consult this review article for this Computer Speech and Language 34 (2015) 87–112. There are probably more than a hundred papers on this in IEEE journals. Why would one do this with MoS₂ transistor? There is not a single advantage highlighted by the authors in the motivation section. What novel property MoS₂ bring to table. If all you need is top-gate electrodes arranged in this design then any semiconductor would do. The proposed circuit is not even well scaled or even faster than the commercial chips.
- 4) The author used verbose language without limits. The reader has to go through the history of sound localization to know what the authors actually did. The coincidence counter by Walther Bothe and Bruno Rossi is used for high energy particles and has no relevance to this topic so why bring it up.
- 5) Finally, the author proposed a multi-terminal neuromorphic device based on MoS₂ and failed to cite the first paper in this emerging new field, Nature 554, 500 (2018).

In conclusion, lack of novelty (point 1), lack of full demonstration (point 2), and lack of any advantage from commercial sound localization chips make this manuscript highly specialized. The authors could reconsider submitting to the Journal of Applied Physics after demonstrating a working circuit.

Reviewer #3 (Remarks to the Author):

Comments:

I have carefully read the manuscript by Sarbashis Das et al, entitled "A Biomimetic 2D Transistor for Audiomorphic Computing". In this manuscript, the authors introduce an interesting solid state biomimetic audiomorphic device that consists of multiple split gated MoS₂ FET for emulating the spatial map of coincidence detector neurons and tunable RC circuits for imitating the time delay neurons following the Jeffress model of sound localization. In this manuscript, a different approach towards neuromorphic computing is proposed here to mimic the animal extraordinary sensory capabilities by embracing the neurobiological architecture and computational map that exist inside the auditory cortex of barn owl. In my opinion, this manuscript can be accepted by Nature Communication if the following issues can be carefully addressed:

1. In Fig. 4b, the authors directly show the results of a 3D plot, but do not seem to explain how the data are obtained. I suggest that relative descriptions should be added to this part.
2. In Fig. 4f, the authors claimed that the resistances of fully top gated MoS₂ FETs connected to the split gates increased monotonically from left (right) to right (left). How was this varying resistance set during the experiment? Only one fully top gated MoS₂ FET was fabricated, or many?
3. On Page 19, what does this sentence "The resistance scaling can be achieved seamlessly by scaling the width of the MoS₂ FETs" mean? I read this part several times but still had a lot of trouble understanding it.
4. In Figure 4g, there is not enough explanation about the provided graph.
5. In figure 5c, the transfer characteristics of the split gated MoS₂ FET with 200 nm spacing are measured under different back gate biases. For comparison, can you include a plot of data such as Fig. 5c for a fully top gated MoS₂ FET to prove your conclusions?

6. Page 10, lines 16: “the red curve” should be changed to “the yellow curve”. Moreover, the horizontal annotation in figure 5g should be “ ΔIRD ”, not “ ΔIRF ”. Similar issues in the manuscript should be carefully checked and revised.

7. For the 2D MoS₂ neuromorphic transistor, there are several recent important references to be missing. For example, Nature Materials, 18, 141-148(2019); Nanoscale, 11, 1360-1369(2019); Journal of Materials Chemistry C, 2019, 682-691(2019); ACS Applied Materials & Interfaces, 10, 25943-25948(2018); Organic Electronics, 63, 120-128(2018). Can you compare your work and the above references in your manuscript? What is your advantage and disadvantage?

In summary, this work is a new finding in the 2D MoS₂ neuromorphic transistor. The above points should be fully addressed before it could be accepted to be a publication in the high-impact journal of Nature Communication. Hope we can see the next beautiful work from your research group!

Reviewer #1 (Remarks to the Author):

The manuscript by Das et al. described a multi-gate MoS₂ FET. Each pair of the gate serves as a NAND for coincidence detection. In addition, the inputs to gates see different RC with different gates, achieved by wiring extra FETs as resistors. Since paired gates are of different gaps, the multi-gate FET produces different analog currents upon different time gaps between the signals, like the Jeffress's model for the sound detection. In addition, back gate modulation is also investigated. The results are interesting and timely. However, there are some questions to be addressed.

We would like to thank the reviewer for acknowledging and appreciating our work. We are also pleased that the reviewer finds the results to be interesting and timely.

First, multi-gate logic has long been studied. The authors need to show the advantages (e.g. footprint, scalability, performance like S/N ratio) that a multi-gate FET possesses over the functional equivalent logic formed by conventional FETs. Say for instance, a NAND of 2 single-gate FETs connected in parallel.

We thank the reviewer for bringing up the relevant issues of footprint, scalability and S/N ratio.

Footprint: The audiomorphic MoS₂ transistor is expected to provide significant area benefit when compared to functionally equivalent logic formed by NAND gates. This is because NAND logic gates in the CMOS technology consist of 4 FETs (2 PMOS and 2 NMOS) each with 3 terminals (source, drain and gate). Therefore, to match N_P split gate pairs the NAND CMOS circuit layout would require an area of $\sim 12 \cdot N_P \cdot A$, where, A is the layout area occupied by a single terminal corresponding to a given technology node. However, the layout for the analog multi-gate audiomorphic transistor would necessitate an area of $\sim (2 \cdot N_P + 2) \cdot A$, which include the N_P split gate pairs and 1 source and 1 drain terminal. Clearly, area benefit becomes significant with

increasing N_P . However, this argument is rather simplistic and requires further in depth analysis. We feel that it will be premature at this stage to derive any meaningful qualitative conclusion on the footprint based on the prototype analog audiomorphic device, which is neither aggressively scaled nor optimally designed.

Scalability: As discussed by Frank *et al.*[1], the scalability of ultra-thin body MOSFET is captured through a simple parameter called the screening length, which is given by the following equation:

$$\lambda_{SC} = \lambda_{geo} = \sqrt{\frac{\epsilon_{body}}{\epsilon_{ox}} t_{body} t_{ox}}$$

Where, t_{body} and t_{ox} are the thicknesses and, ϵ_{body} and ϵ_{ox} are the dielectric constants of the channel and the gate oxide, respectively. In order to avoid short channel effects the channel length of an FET (L_{CH}) has to be at least three times higher than the screening length, i.e. $L_{CH} > 3\lambda_{SC}$. For atomically thin semiconducting monolayers of 2D materials, $t_{body} \approx 0.6$ nm, which corresponds to $\lambda_{SC} \approx 1.3$ nm, whereas, for the most advanced FinFET technology the thickness of Si fin can be scaled down to only 5 nm without severely increasing the bandgap due to quantum confinement effects and reducing the mobility due to enhanced surface roughness scattering [2]. Therefore, use of 2D material such as MoS₂ allows for the geometric miniaturization without any loss of electrostatic integrity. In other word more split gate pairs can be fitted into the audiomorphic device for a given channel length of the MoS₂ FET.

S/N Ratio: Fig. 3g can be used to understand the noise tolerance of the audiomorphic device. Note that the number of split gate pairs or coincidence neurons, N_P , determines the number of analog computation levels which in turn determines the angular precision, $\Delta\theta$, of the biomimetic audiomorphic device following the relationship, $\Delta\theta = 180^\circ / N_P$. From this expression, it may appear that the angular precision can be made arbitrarily small by increasing the number of analog levels. However, a large number of analog levels would require a wider analog range for the current suppression or IR or in other words, it necessitates a semiconducting channel with large current ON/OFF ratio. If the minimum required difference between the analog IR values corresponding to two consecutive split gate pairs is ΔIR_D , then N_P and hence $\Delta\theta$ will be determined by the following expression.

$$N_P = \frac{I_{ON}/I_{OFF}}{\Delta I R_D}; \quad \Delta\theta = 180^\circ \frac{\Delta I R_D}{I_{ON}/I_{OFF}} \quad [2]$$

Fig. 3g shows $\Delta\theta$ and N_P as a function of $\Delta I R_D$ for different values of I_{ON}/I_{OFF} . Note that a small value for $\Delta I R_D$ results in greater precision but also requires larger number of split gate pairs. Furthermore, small $\Delta I R_D$ can be more susceptible to noise. Interestingly, the same precision can be achieved for a larger $\Delta I R_D$ by increasing the ON/OFF ratio.

Second, the detection in Fig. 3 is based on analog I_{ds} currents, which demands extra components like ADCs that are not counted here. The authors may like to evaluate the system level performance and compare that with conventional logic circuits (e.g. multiple NANDs sharing common inputs. Each NAND has its dedicated output, rather than a shared common analog output).

The reviewer brings up an excellent point. We indeed agree with the reviewer that it is important to evaluate the system level performance of the audiomorphic device and compare it with fully integrated conventional logic circuits e.g. multiple NANDs sharing common inputs. However, kindly consider the fact that the purpose of this paper is to introduce and experimentally demonstrate a new paradigm of neuromorphic computing, which we refer to as biomimetic computing, using a relatively simple split-gated field-effect transistor (SGFET) device coupled with some delay units. Therefore, we feel that full system level evaluation will be premature and far-fetched at this early stage where we attempt to demonstrate a proof-of-concept structure. We would certainly focus on the issues concerning the peripheral circuits like ADCs in our future work for the system level integration.

Third, both the abstract and the main text are relatively lengthy, with probably too much background information. The referee agrees that the background information is mandatory, but too much background may dilute the focus. The referee suggests making the writing concise, with introduction tightly related to the background.

We agree with the reviewer.

We have revised the abstract and introduction to make them concise and tightly related to the background.

The detailed technical comments are as following.

1. The abstract could be more concise to highlight what is the primary novelty, and what's the secondary.

We have revised the abstract.

2. The introduction comprises psychological claims that lack references. The referee doubts whether it is valid to claim human learning is driven by curiosity while that of animals is not. In addition, the referee tends to feel that human behaviors are essentially still demands driven, but humans could foresee the long-term consequences of actions. In short, references are needed for claims. And as mentioned, the introduction should be more closely related to the main achievement, which is the simulation of Jeffress's model.

We agree with the reviewer that it might not be valid to make the claim that human learning is driven by curiosity while that of animals is not. We also concur with the reviewer's opinion that human behaviors are essentially still demands driven, but humans could foresee the long-term consequences of actions.

We have removed these claims from the introduction section and made it more concise in the revised manuscript.

3. The referee wonders, what are the advantages of using HSQ as a gate oxide? Could HSQ make a good interface with the MoS₂? What is the mobility difference between the FET with HSQ gate and that with back gate alone?

The reviewer has raised an excellent point. In our recent publication (Nasr, J. R. *et. al* Advanced Electronic Materials, 5(4), 1800888, 2019), we have discussed in detail the advantage of using

hydrogen silsesquioxane (HSQ) as the gate dielectric for MoS₂. In short, HSQ is a negative tone electron beam (e-beam) resist with the capability of achieving well-defined features as small as 6 nm [3]. HSQ is extensively used as microelectronic etch mask and inter-metal dielectric for integrated circuits within the semiconductor industry [4]. In addition, the thickness of HSQ can be scaled down to below 10 nm by optimizing the spin-speed, solubility, and viscosity [3]. Furthermore, HSQ requires low processing temperatures, and most importantly, it possesses dielectric properties similar to SiO₂ when exposed to high-dose of e-beam irradiation and thermal curing in inert atmospheres. Therefore, HSQ offers a facile, low-temperature, scalable and universally applicable fabrication scheme for dual-gated 2D-FETs, which is compatible with the back-end-of-the-line (BEOL) process flow of complementary metal oxide semiconductor (CMOS) technology.

Regarding the mobility difference between the FET with HSQ gate and that with back gate alone, please consider Fig. R1. Fig R1a and Fig R1b, respectively, show the dual sweep back gated transfer characteristics of MoS₂ FET before and after the fabrication of the HSQ top gate and Fig R1c and Fig R1d, respectively, show the corresponding mobility values as a function of the back gate voltage. We found none to minimal changes in the device performance as evident from both the forward and reverse sweep threshold voltage (V_T), and carrier (electron) mobility values, tabulated in Fig R1e. In fact, the hysteresis window was found to improve significantly from ~9V to ~3V after the HSQ top gate fabrication since the HSQ also acts as a capping layer for the MoS₂ channel and helps to keep the adsorbents away from influencing carrier trapping at the interface.

Figure R1. Dual sweep back gated transfer characteristics of MoS₂ FET a) before and b) after the fabrication of the HSQ top gate. Corresponding mobility values as a function of the back gate voltage c) before and d) after the fabrication of the HSQ top gate. e) Table showing threshold voltage for both forward and reverse sweep, extracted carrier (electron) mobility from transconductance and hysteresis window.

4. As mentioned in the general comment, the advantages of the multi-gate FET need to be pointed out. On the contrary, the logic equivalent NAND with a pair of single-gate FETs could be benefited with less gate coupling and less fringing electric field.

The audiomorphic MoS₂ transistor is expected to provide significant area benefit when compared to functionally equivalent logic formed by NAND gates. This is because NAND logic gates in the CMOS technology consist of 4 FETs (2 PMOS and 2 NMOS) each with 3 terminals (source, drain and gate). Therefore, the CMOS circuit layout to match N_P split gate pairs would require an area of $\sim 12 \cdot N_P \cdot A$, where, A is the layout area occupied by a single terminal corresponding to a given technology node. However, the layout for the analog multi-gate audiomorphic transistor would necessitate an area of $\sim (2 \cdot N_P + 2) \cdot A$, which include the N_P split gate pairs and 1 source and 1 drain terminal. Clearly, area benefit becomes significant with increasing N_P. However, this argument is rather simplistic and requires further in depth analysis. We genuinely feel that it is premature to derive any meaningful qualitative conclusion based on the prototype analog audiomorphic device, which is neither aggressively scaled nor optimally designed to minimize gate coupling and effect of fringing electric field.

5. What is the temporal width of the voltage pulses in Fig. 2g? How fast could this logic gate switch?

In Fig. 2g, the temporal width of the voltage pulses is 10 ms. Note that there is no fundamental difference between our split gated MoS₂ FET and a conventional dual gated metal-oxide-semiconductor field effect transistor (MOSFET). Therefore, the intrinsic switching speed can be as fast as any other high performance MOSFET, if the device is scaled accordingly. In fact, MoS₂ FETs operating at Gigahertz Frequencies have already been demonstrated [5, 6]. However, in the present demonstration, we are limited by parasitic capacitances due to the use of large metal pads and global back gate configuration. Furthermore, our devices are not scaled properly. One of the advantages of using ultra-thin body MoS₂ as the semiconducting channel material is the better electrostatic gate control that allows aggressive scaling for the device that aids faster switching.

We have added the information regarding the width of the voltage pulses in the revised manuscript.

6. The referee suggests combining Fig. 2j with the experimental curve and making discussion concise, unless the simulation results are related to the novelty of this work.

Reviewer's suggestion is noted. It is indeed possible to combine Fig. 2j with Fig. 2e. However, we feel that the flow of the manuscript would be served better if we discuss the simulation results in a different figure. To the best of our knowledge, there is no precedence where virtual source (VS) model and COMSOL simulations have been combined to explain the performance of split-gated 2D FETs. Therefore, in a way our approach is novel.

We have included the following statement in the abstract to indicate that our approach of combining virtual source (VS) model with COMSOL simulations is a secondary novelty of the manuscript:

Finally, we have combined the virtual source (VS) model that describes current transport through nano devices with finite element COMSOL multiphysics simulations to explain and project the performance of our biomimetic audiomorphic transistor.

7. It will be more helpful to highlight the MoS₂ flake in Fig. 3a or b.

We agree with the reviewer. However, please note that it is very difficult to image such an ultra-thin MoS₂ flake once the split gate stack with 120 nm HSQ has been fabricated. Therefore, we have marked the boundary of the flake based on the exfoliation prior to the top gate fabrication.

We have marked the boundary of the MoS₂ flake in Fig. 3b.

8. Regarding the Fig. 4c-e, is this two-transistor scheme realized on the same chip? Or these devices are wired via probes?

Thanks for the question. We would like to clarify that for Fig. 4c-e, the two-transistor scheme was indeed realized on the same chip.

9. How are the capacitance ($180 \pm 20\text{pF}$) of the different gates measured? Why the gate capacitances dose not scale with the gap distance of paired gates given the back gate is just 285 nm?

Reviewer's observation is spot-on. In an ideal scenario, the gate capacitance should scale with the gap distance between the split gate pairs. However, in our device structure each of the split gates is connected to a large metal pad so that we can probe them for electrical measurements. The area of the actual split gate ($\sim 0.5 \times 10 \mu\text{m}^2$) is negligible compared the pad area ($75 \times 75 \mu\text{m}^2$). Therefore, the capacitance value extracted from our delay measurements are mostly dominated by parasitic pad capacitances.

10. For Fig. 4f, what are the advantages of using FETs to cause RC delay rather than just resistors? Does the delay need to be tuned?

The reviewer is correct in his assessment that simple resistors can be also be used to cause the RC delays. However, the use of FET has the advantage that the resistance value can be tuned dynamically without changing the dimensions of the device since FETs are ultimately voltage controlled variable resistors. Furthermore, note that according to Fig. 1a, the interaural time delay (ITD) depends on the head size (r_H), which in this case is the distance between the audio transducers and the velocity of sound wave (v_S). Both are subjected to small variations when deployed for practical usage. Therefore, adding the capability of tuning the delay on-chip will make the design much more robust.

11. It may be better to include the original waveforms in Fig. 4g, in addition to indications of the places where peak voltages are reached.

We agree with the reviewer.

Fig. 4g was revised to add the original voltage waveforms along with the places where peak voltages are reached.

12. Usually neuroplasticity is due to synaptic plasticity that changes the connection strengths between neurons. Is it proper to term back gate effect as an analog of the neuroplasticity?

The reviewer brings up a legitimate point. The neuroplasticity is defined as the ability of the brain to change the weights of the synapses or the connections between the neurons in response to learning stimuli from the external world. One of the examples of this neuroplasticity is when the brain adjusts the sensory inputs to focus attention on a certain task. For example, in humans, when a person is sleeping, the brain inhibits the ambient sounds so that the sleep is not disturbed. The same person when driving is alert and the brain is receiving and processing all the ambient sounds. This is exactly what we have tried to mimic by varying the back-gate voltage. We believe that our demonstration of neuroplasticity is a specific example out of the countless ways in which the brain uses neuroplasticity.

Reviewer #2 (Remarks to the Author):

This manuscript by Das et al. demonstrates an electronic device based on a 2D semiconductor MoS₂ to mimic the spatiotemporal responses needed for sound localization. The device consists of a MoS₂ field-effect transistor with multiple top-gate electrodes. The top gates are fabricated in pairs with varying gaps such that local conductivity of the channel is controlled by simple electrostatic control in the gap. The field is topical. However, as listed below, several serious problems with this manuscript make it unsuitable for Nature Communications.

1) Showing sound localization by spatiotemporal responses in MoS₂ transistors has been shown before in this paper Nano Lett., 2018, 18 (5), 3229. The authors do not acknowledge this paper or even cite it. So, the major claim of novelty has already been taken and ignoring that fact is not helping here.

The author brings forth a very interesting paper in this topical field. The paper Nano Lett., 2018, 18 (5), 3229, demonstrates spike time dependent plasticity (STDP) implemented using Joule heating in monolayer MoS₂ which is subsequently used to demonstrate sound localization using a spiking neural network (SNN). They use excitatory and inhibitory synaptic devices for detecting an interaural time difference (ITD) by suppressing sound intensity- or frequency-dependent synaptic connectivity. While the concept and the results are certainly very interesting, they are markedly different from our work. The novelty of our approach arises for the fact that we do not use the generic structure of artificial neural network (ANN) that involves only the synaptic strength (weights for conventional ANN or STDP for SNN) and neurons (activation function). Instead, we have captured the specific neurobiological architectures and associated computational algorithms that exist inside the brainstem of animals with superior sensory capabilities. As an example, we study barn owl – a nocturnal animal known for its extraordinary hunting capability in complete darkness using sound cues with the precision of 1-2⁰. We learned that for typical size of animal heads the ITDs are in the range of hundreds of microseconds which must be processed by neurons which can spike only once per few milliseconds. Therefore, auditory signal processing is a challenging computational task that requires special neurobiological algorithm and architecture compared to a simple SNN. Famous American scientists Jeffress formulated a model that describes

how ITDs are represented as a “place” in an array of nerve cells or in other words how the brain transforms temporal spike coding into spatial coding [7]. Remarkably, the key assumptions of his model are valid even today and the model remains as the cornerstone for the neurophysiological understanding and development of virtually any computational models for sound localization [8, 9]. We, therefore, implemented the Jeffress model of sound localization in hardware that involves artificial coincidence detector neuron, time delay neuron and their spatial organization or map. Any of these neurons can be realized using spiking neuron as well, but the basic philosophy is completely different from a traditional SNN approach. We certainly appreciate the work presented in the paper Nano Lett., 2018, 18 (5), 3229, however, we could not find any reference to the spatial neural map or Jeffress model, which are fundamental to sound localization in many species. Furthermore, our paper does not use Joule heating as the working mechanism for synaptic transmission. Instead, the neural coincidence concept is implemented using split gated transistor geometry and the spatial map for the coincidence is realized by changing the gap between the split gate pairs, which changes the strength of fringing field in the un-gated region of the MoS₂ FET and thereby the current conduction through the device. In the light of the above discussion, we are confident that our approach is novel and has none to minimal overlap with the paper pointed out by the reviewer.

We have added the missing reference in the revised manuscript.

2) Although the authors show a tremendous amount of data in the paper the full sound localization operation is actually not demonstrated. Physically, two different neurons are needed -coincidence neurons and time-delay neuron. The authors show the operation of coincidence neuron in much details in Fig. 1 - 3. The individual time-delay neuron responses are shown in Fig. 4d, e. But I am not seeing the integration of the two for sound localization. There is a schematic in Fig. 4f but did they actually make the circuit? If so, the experimental data should be presented.

Reviewer’s observation is correct. We have not integrated the coincidence neurons with their corresponding time delay neurons for the demonstration of full sound localization operation. The schematic in Fig. 4f is shown as a guide to this integration. While we agree that such integration

is important, in the context of the paper it will hardly add any value. The purpose of this paper is two-folded. First, we introduce a new paradigm of neuromorphic computing, which is referred to as biomimetic computing that draws inspiration from specific neural computational algorithms implemented using befitting neurobiological organization of neurons inside the brainstem of natural super sensory animals. Second, we experimentally demonstrate how novel devices, such as split-gated field-effect transistors coupled with relatively simple delay units can accomplish the same. We would certainly be interested in full system level integration in our future follow-up research.

3) Another related issue is why anyone would want to do sound localization with MoS₂ transistors? First of all, there are quite advanced silicon CMOS chips which are used in commercial robotics and navigation system. The authors may consult this review article for this Computer Speech and Language 34 (2015) 87–112. There are probably more than a hundred papers on this in IEEE journals. Why would one do this with MoS₂ transistor? There is not a single advantage highlighted by the authors in the motivation section. What novel property MoS₂ bring to table. If all you need is top-gate electrodes arranged in this design then any semiconductor would do. The proposed circuit is not even well scaled or even faster than the commercial chips.

The reviewer does bring up a valid point regarding the use of MoS₂ FETs for sound localization. We do agree with the reviewer that our approach is not restricted to the use of MoS₂ as the semiconducting channel material. The concept of audiomorphic computing based on the Jeffress model can in fact be implemented using any planar semiconducting material, which allows the implementation of the split gate geometry. However, an argument can be made that the atomically thin body nature of the 2D semiconductors allows for better electrostatic gate control, whereas optical transparency and mechanical flexibility makes them compatible with various emerging Internet of Things (IoT) applications. We also agree with the reviewer that there are advanced silicon CMOS chips, which are used in commercial robotics and navigation system. However, our work is still unique in the fact that we are able to mimic the true biological architecture and the associated neural algorithm for sound localization using a rather simple device. It is true that our audiomorphic device is neither aggressively scaled nor superfast. However, please note that the

purpose of this article is to introduce the first proof-of-concept experimental demonstration of a new paradigm of neuromorphic computing, namely biomimetic computing.

We have added the following discussion regarding our motivation behind the use of MoS₂ and pointed out alternate choices for implementing the biomimetic neuromorphic device:

Note that our choice of MoS₂ as the semiconducting channel material is motivated by the increasing interest in two-dimensional (2D) layered materials as the successor for the aging Si technology [10-12]. The atomically thin body nature of these 2D semiconductors not only allow for aggressive channel length scaling *via* superior gate electrostatics but also the transparent and flexible aspects make them appealing for various emerging technologies such as the Internet of Things (IoT).[13-16] Furthermore, recent years have witnessed excellent progress in neuromorphic computing based on MoS₂ devices.[17-24] However, the concept of biomimetic neuromorphic computing need not be restricted to MoS₂ FETs. In fact, it can be implemented using any semiconducting material, which allows for the implementation of the split gate FET geometry.

4) The author used verbose language without limits. The reader has to go through the history of sound localization to know what the authors actually did. The coincidence counter by Walther Bothe and Bruno Rossi is used for high-energy particles and has no relevance to this topic so why bring it up.

We are sorry if the reviewer felt that way. We agree that coincidence counter by Walther Bothe and Bruno Rossi is used for high-energy particles and has no relevance to this topic.

We have removed the section and revised the language in the abstract and introduction.

5) Finally, the author proposed a multi-terminal neuromorphic device based on MoS₂ and failed to cite the first paper in this emerging new field, Nature 554, 500 (2018).

We would like to thank the reviewer for pointing out to this missing reference.

We have added these missing references in the revised manuscript.

In conclusion, lack of novelty (point 1), lack of full demonstration (point 2), and lack of any advantage from commercial sound localization chips make this manuscript highly specialized. The authors could reconsider submitting to the Journal of Applied Physics after demonstrating a working circuit.

We hope we have answered the above concerns in response to the specific comments raised by the reviewer. We are surprised to see such negativity and strong opposition by the reviewer.

Reviewer #3 (Remarks to the Author):

Comments:

I have carefully read the manuscript by Sarbashis Das et al, entitled “A Biomimetic 2D Transistor for Audiomorphic Computing”. In this manuscript, the authors introduce an interesting solid state biomimetic audiomorphic device that consists of multiple split gated MoS2 FET for emulating the spatial map of coincidence detector neurons and tunable RC circuits for imitating the time delay neurons following the Jeffress model of sound localization. In this manuscript, a different approach towards neuromorphic computing is proposed here to mimic the animal extraordinary sensory capabilities by embracing the neurobiological architecture and computational map that exist inside the auditory cortex of barn owl. In my opinion, this manuscript can be accepted by Nature Communication if the following issues can be carefully addressed:

We would like to thank the reviewer for the appreciation of our work. We have addressed the reviewer’s concern in the revised manuscript.

1. In Fig. 4b, the authors directly show the results of a 3D plot, but do not seem to explain how the data are obtained. I suggest that relative descriptions should be added to this part.

We thank the reviewer for pointing out the omission.

The following description has been added to the revised manuscript:

Fig. 4b shows a 3D plot for the length of delay line (L_D) required for mapping the ITDs (t_{ITD}) as a function of the signal propagation velocity (v_p) following the expression $L_D = v_p t_{ITD}$. The red circle indicates that the length of metallic interconnect that is necessary to map ITDs that are in the range of 100 μ s is \sim 1m. This is impractical for any integrated circuit and arises from the fact that the speed of electronic signal transmission through metal is relatively fast at $\sim 10^6$ m/s. On the contrary, as indicated by the green circle, the barn owl accomplishes the same by using an intermodal distance of only $\sim 60 \mu$ m [25] for the axons that act as delay lines within the NL. This is simply due to the orders of magnitude slower signal propagation through the axons as discussed

earlier. Therefore, it is necessary to introduce solid-state delay elements into our biomimetic device in order to capture the time scale of axonal conduction and thereby reduce the required wire length.

2. In Fig. 4f, the authors claimed that the resistances of fully top gated MoS₂ FETs connected to the split gates increased monotonically from left (right) to right (left). How was this varying resistance set during the experiment? Only one fully top gated MoS₂ FET was fabricated, or many?

The reviewer is correct. Only one fully top-gated MoS₂ FET was fabricated for the proof-of-concept demonstration of artificial time delay neuron. We have not integrated the coincidence neurons with their corresponding time delay neurons for the demonstration of full sound localization operation. The schematic in Fig. 4f is shown as a guide to this integration. While we agree that such integration is important, in the context of the paper it will hardly add any value. The purpose of this paper is two-folded. First, we introduce a new paradigm of neuromorphic computing, which is referred to as biomimetic computing that draws inspiration from specific neural computational algorithms implemented using befitting neurobiological organization of neurons inside the brainstem of natural super sensory animals. Second, we experimentally demonstrate how novel devices, such as split-gated field-effect transistors coupled with relatively simple delay units can accomplish the same. We would certainly be interested in full system level integration in our future follow-up research.

3. On Page 19, what does this sentence “The resistance scaling can be achieved seamlessly by scaling the width of the MoS₂ FETs” mean? I read this part several times but still had a lot of trouble understanding it.

We are sorry for the confusion. Please note that in order to realize the audiomorphic circuit each split gate must be connected to a delay element with unique time delay determined by the ITDs. In our demonstration, these delay elements are realized using simple RC circuits. As shown in Fig. 4f, we use the channel resistance of the MoS₂ FETs denoted by R₁-R₅ connected to the corresponding split gates, which provide the capacitances *via* their gate capacitances to achieve those unique delays. The channel resistance of the MoS₂ FETs depend on the channel conductance

and the dimension of the channel i.e. the length and the width. The channel conductance can be tuned by applying different gate biases and this is the approach we adopt to demonstrate different time constants in our proof-of-concept artificial time delay neuron in Fig. 4c-e. However, in an integrated circuit one would like to avoid applying different gate voltages to different MoS₂ FETs. Therefore, a much favorable approach will be to design and fabricate MoS₂ FETs with different length and width to match the required resistance values. We agree with the reviewer that the term “scaling” is probably not the right phrase to use in this context and can mislead the reader.

We have revised the sentence as follows:

The required resistance values can be achieved by designing the length and width of the MoS₂ FETs accordingly.

4. In Figure 4g, there is not enough explanation about the provided graph.

We thank the reviewer for pointing out the missing explanation. Please note that we have now revised Fig. 4g to add the original voltage waveforms along with the places where peak voltages are reached for different ITDs. We have also added the explanation in the text.

Fig. 4g was revised to add the original voltage waveforms along with the places where peak voltages are reached. The following discussion was introduced in the revised manuscript:

Fig. 4g shows how each split gate is charged when the acoustic transducers at the two ends of the chip receive the sound signal with positive, zero and negative interaural delays (ITDs). The spikes mark the times when the respective gates reach their peak negative potential. Color coding shows that the coincidence occurs at the central split gate (green) for ITD = 0, and on the right side (yellow) for positive ITD and left side (red) for negative ITD.

5. In figure 5c, the transfer characteristics of the split gated MoS₂ FET with 200 nm spacing are measured under different back gate biases. For comparison, can you include a plot of data such as Fig. 5c for a fully top-gated MoS₂ FET to prove your conclusions?

This is an excellent suggestion by the reviewer.

We have now included the transfer characteristics of fully top-gated MoS₂ FET for different back gate voltages in the supporting information section of the manuscript.

Figure R2. Transfer characteristics of fully top-gated MoS₂ FET for different back gate voltages.

6. Page 10, lines 16: “the red curve” should be changed to “the yellow curve”. Moreover, the horizontal annotation in figure 5g should be “ΔIRD”, not “ΔIRF”. Similar issues in the manuscript should be carefully checked and revised.

Thank you for spotting these inconsistencies.

We have fixed the coloring and labeling and carefully checked the manuscript.

7. For the 2D MoS₂ neuromorphic transistor, there are several recent important references to be missing. For example, Nature Materials, 18, 141-148(2019); Nanoscale, 11, 1360-1369(2019); Journal of Materials Chemistry C, 2019, 682-691(2019); ACS Applied Materials & Interfaces, 10, 25943-25948(2018); Organic Electronics, 63, 120-128(2018). Can you compare your work and the above references in your manuscript? What is your advantage and disadvantage?

We would like to thank the reviewer for pointing out to these missing references. These are indeed

excellent exhibition of neuromorphic computing using MoS₂ based devices. Note that the difference lies in the fact that these demonstrations derive their inspiration from the generic nature of synaptic connection between the neurons inside the mammalian brain for hardware realization of artificial neural networks (ANNs), whereas, we focus on capturing the specific neurobiological architectures and associated computational algorithms that exist inside the brainstem of animals with extraordinary sensory capabilities. We think that both approaches are useful and important for mimicking the brain and thereby advance the field of neuromorphic computing and artificial intelligence.

We have added these missing references in the revised manuscript.

In summary, this work is a new finding in the 2D MoS₂ neuromorphic transistor. The above points should be fully addressed before it could be accepted to be a publication in the high-impact journal of Nature Communication. Hope we can see the next beautiful work from your research group!

We are thankful to the reviewer for their kind words and appreciation of our research.

References

- [1] D. J. Frank, Y. Taur, and H.-S. P. Wong, "Generalized scale length for two-dimensional effects in MOSFETs," *Electron Device Letters, IEEE*, vol. 19, pp. 385-387, 1998.
- [2] E. Yu, L. Chang, S. Ahmed, H. Wang, S. Bell, C.-Y. Yang, *et al.*, "FinFET scaling to 10 nm gate length," in *Electron Devices Meeting, 2002. IEDM'02. International, 2002*, pp. 251-254.
- [3] A. E. Grigorescu, M. C. van der Krogt, C. W. Hagen, and P. Kruit, "10nm lines and spaces written in HSQ, using electron beam lithography," *Microelectronic Engineering*, vol. 84, pp. 822-824, 2007.
- [4] Y. Y. Chen, S. M. Jang, C. H. Yu, S. C. Sun, and M. S. Liang, "A high performance and reliable low-k inter-metal dielectric using hydrogen silsesquioxane (HSQ)," *Proceedings of the IEEE 1999 International Interconnect Technology Conference, 1999*.
- [5] H. Y. Chang, M. N. Yogeesh, R. Ghosh, A. Rai, A. Sanne, S. Yang, *et al.*, "Large-Area Monolayer MoS₂ for Flexible Low-Power RF Nanoelectronics in the GHz Regime," *Adv Mater*, vol. 28, pp. 1818-23, Mar 02 2016.
- [6] D. Krasnozhan, D. Lembke, C. Nyffeler, Y. Leblebici, and A. Kis, "MoS₂ Transistors Operating at Gigahertz Frequencies," *Nano Letters*, vol. 14, pp. 5905-5911, 2014/10/08 2014.
- [7] L. A. Jeffress, "A place theory of sound localization," *Journal of comparative and physiological psychology*, vol. 41, p. 35, 1948.
- [8] P. X. Joris, P. H. Smith, and T. C. Yin, "Coincidence detection in the auditory system: 50 years after Jeffress," *Neuron*, vol. 21, pp. 1235-1238, 1998.
- [9] L. O. Trussell, "Synaptic mechanisms for coding timing in auditory neurons," *Annual review of physiology*, vol. 61, pp. 477-496, 1999.
- [10] S. Das, J. A. Robinson, M. Dubey, H. Terrones, and M. Terrones, "Beyond Graphene: Progress in Novel Two-Dimensional Materials and van der Waals Solids," *Annual Review of Materials Research, Vol 45*, vol. 45, pp. 1-27, 2015.
- [11] G. R. Bhimanapati, Z. Lin, V. Meunier, Y. Jung, J. Cha, S. Das, *et al.*, "Recent Advances in Two-Dimensional Materials beyond Graphene," *ACS Nano*, vol. 9, pp. 11509-11539, 2015/12/22 2015.
- [12] B. Radisavljevic, M. B. Whitwick, and A. Kis, "Integrated circuits and logic operations based on single-layer MoS₂," *ACS Nano*, vol. 5, pp. 9934-8, Dec 27 2011.
- [13] S. Das, R. Gulotty, A. V. Sumant, and A. Roelofs, "All two-dimensional, flexible, transparent, and thinnest thin film transistor," *Nano Lett*, vol. 14, pp. 2861-6, May 14 2014.
- [14] A. J. Arnold, A. Razavieh, J. R. Nasr, D. S. Schulman, C. M. Eichfeld, and S. Das, "Mimicking Neurotransmitter Release in Chemical Synapses via Hysteresis Engineering in MoS₂ Transistors," *ACS nano*, vol. 11, pp. 3110-3118, 2017.
- [15] D. S. Schulman, A. Sebastian, D. Buzzell, Y.-T. Huang, A. J. Arnold, and S. Das, "Facile Electrochemical Synthesis of 2D Monolayers for High-Performance Thin-Film Transistors," *ACS Applied Materials & Interfaces*, vol. 9, pp. 44617-44624, 2017/12/27 2017.
- [16] S. Das, "Two Dimensional Electrostrictive Field Effect Transistor (2D-EFET): A sub-60mV/decade Steep Slope Device with High ON current," *Sci Rep*, vol. 6, p. 34811, Oct 10 2016.
- [17] D. Xie, W. Hu, and J. Jiang, "Bidirectionally-triggered 2D MoS₂ synapse through coplanar-gate electric-double-layer polymer coupling for neuromorphic complementary spatiotemporal learning," *Organic Electronics*, vol. 63, pp. 120-128, 2018.
- [18] D. Xie, J. Jiang, W. Hu, Y. He, J. Yang, J. He, *et al.*, "Coplanar Multigate MoS₂ Electric-Double-Layer Transistors for Neuromorphic Visual Recognition," *ACS Applied Materials & Interfaces*, vol. 10, pp. 25943-25948, 2018/08/08 2018.

- [19] W. Hu, J. Jiang, D. Xie, B. Liu, J. Yang, and J. He, "Proton–electron-coupled MoS₂ synaptic transistors with a natural renewable biopolymer neurotransmitter for brain-inspired neuromorphic learning," *Journal of Materials Chemistry C*, vol. 7, pp. 682-691, 2019.
- [20] J. Jiang, W. Hu, D. Xie, J. Yang, J. He, Y. Gao, *et al.*, "2D electric-double-layer phototransistor for photoelectronic and spatiotemporal hybrid neuromorphic integration," *Nanoscale*, vol. 11, pp. 1360-1369, 2019.
- [21] X. Zhu, D. Li, X. Liang, and W. D. Lu, "Ionic modulation and ionic coupling effects in MoS₂ devices for neuromorphic computing," *Nature Materials*, vol. 18, pp. 141-148, 2019/02/01 2019.
- [22] V. K. Sangwan, D. Jariwala, I. S. Kim, K.-S. Chen, T. J. Marks, L. J. Lauhon, *et al.*, "Gate-tunable memristive phenomena mediated by grain boundaries in single-layer MoS₂," *Nature nanotechnology*, vol. 10, p. 403, 2015.
- [23] V. K. Sangwan, H. S. Lee, H. Bergeron, I. Balla, M. E. Beck, K. S. Chen, *et al.*, "Multi-terminal memtransistors from polycrystalline monolayer molybdenum disulfide," *Nature*, vol. 554, pp. 500-504, Feb 21 2018.
- [24] L. Sun, Y. Zhang, G. Hwang, J. Jiang, D. Kim, Y. A. Eshete, *et al.*, "Synaptic Computation Enabled by Joule Heating of Single-Layered Semiconductors for Sound Localization," *Nano Letters*, vol. 18, pp. 3229-3234, 2018/05/09 2018.
- [25] C. Carr and M. Konishi, "A circuit for detection of interaural time differences in the brain stem of the barn owl," *Journal of Neuroscience*, vol. 10, pp. 3227-3246, 1990.

REVIEWERS' COMMENTS:

Reviewer #1 (Remarks to the Author):

The manuscript is in a better shape after the revision, of improved clarity. The literature survey has been enriched as suggested by the other referees. I thank the authors for the detailed analysis by making comparisons with alternative implementations, such as those by transistors. I understand the concerns of the authors but I guess it may be helpful to include part of the performance comparison or projection to either the body text or the SI, which could help to justify the practical value of this novel 2D device. I guess it's also one of the concerns of the second referee. But this is optional and does not demand a future revision. I would like to recommend this manuscript for publication based on the novelty and significance of implementing the neuromorphic functionality with 2D devices.

Reviewer #3 (Remarks to the Author):

The authors have well answered my questions and it can be accepted now.

Reviewer #1 (Remarks to the Author):

The manuscript is in a better shape after the revision, of improved clarity. The literature survey has been enriched as suggested by the other referees. I thank the authors for the detailed analysis by making comparisons with alternative implementations, such as those by transistors. I understand the concerns of the authors but I guess it may be helpful to include part of the performance comparison or projection to either the body text or the SI, which could help to justify the practical value of this novel 2D device. I guess it's also one of the concerns of the second referee. But this is optional and does not demand a future revision. I would like to recommend this manuscript for publication based on the novelty and significance of implementing the neuromorphic functionality with 2D devices.

We would like to thank the reviewer for recommending publication of our manuscript. We also concur with the reviewer that it will be helpful for the readers if we include the performance comparison or projection for the biomimetics device in the supplementary information (SI).

We have included the performance comparison or projection for the biomimetics device in the supplementary information (SI).

Reviewer #3 (Remarks to the Author):

The authors have well answered my questions and it can be accepted now.

We would like to thank the reviewer for recommending acceptance of our manuscript.